# A Correlation-Based Sensing Scheme for Outlier Detection in Cognitive Radio Networks

Muhammad Sajjad Khan [1], Mohammad Faisal [2], Su Min Kim [1], Saeed Ahmed [3], Marc St-Hilaire [4] and Junsu Kim [1],*

1   Department of Electronics Engineering, Korea Polytechnic University, Gyeounggi-do 15073, Korea; sajjad.khan@iiu.edu.pk (M.S.K.); suminkim@kpu.ac.kr (S.M.K.)
2   Department of Computer Science, University of Malakand, Chakdara 18800, Pakistan; mfaisal@uom.edu.pk
3   Department of Electrical Engineering, Mirpur University of Science and Technology, Mirpur 10250, Pakistan; saeed.ahmed@must.edu.pk
4   Department of System and Computer Engineering, Carleton University, Ottawa, ON K1S 5B6, Canada; marc_st_hilaire@carleton.ca
*   Correspondence: junsukim@kpu.ac.kr; Tel.: +82-31-8041-0497

**Abstract:** Cooperative spectrum sensing (CSS) is a vital part of cognitive radio networks, which ensures the existence of the primary user (PU) in the network. However, the presence of malicious users (MUs) highly degrades the performance of the system. In the proposed scheme, each secondary user (SU) reports to the fusion center (FC) with a hard decision of the sensing energy to indicate the existence of the PU. The main contribution of this work deals with MU attacks, specifically spectrum sensing data falsification (SSDF) attacks. In this paper, we propose a correlation-based approach to differentiate between the SUs and the outliers by determining the sensing of each SU, and the average value of sensing information with other SUs, to predict the SSDF attack in the system. The FC determines the abnormality of a SU by determining the similarity for each SU with the remaining SUs by following the proposed scheme and declares the SU as an outlier using the box-whisker plot. The effectiveness of the proposed scheme was demonstrated through simulations.

**Keywords:** cognitive radio networks; spectrum sensing data falsification (SSDF); opposite malicious user (OMU); random opposite malicious user (ROMU); box-whisker plot

## 1. Introduction

The services provided for the rapid growth of the applications, such as computers, laptops, ipads, internet of things (IoT), etc., have increased the demand of the spectrum, which results in spectrum shortage. According to the Federal Communication Commission (FCC), most of the spectrum is underutilized, even in the crowded region, the spectrum utilization is between 15% and 85% [1]. To tackle the issue of the spectrum underutilization, cognitive radio technology (CRT) has emerged as a strong candidate for exploiting the spectrum [2]. The main functionality of CRT is to determine the availability of the spectrum for the secondary users (SUs). For achieving the spectrum availability and efficient utilization of the spectrum, the SUs need to continuously monitor the available spectrum to find the spectrum holes and vacate the spectrum whenever the primary user (PU) appears in the network [3].

The spectrum sensing is performed to determine the presence or absence of the PU in the network. Different detection techniques, such as matched filter detection, cyclostationary feature detection, energy detection, etc., have been proposed in the literature [4,5]. Every detection technique has their own features. For example, feature detection is optimal, when the PU information is available at the SU. When the SU has no prior information about the PU, then the energy detection technique is the optimal detection technique. The SU has only knowledge about the local noise power. The received signal energy is utilized to decide about the existence of the PU in the network.

Various techniques are used to efficiently utilize the scarce spectrum by merging the underlay and overlay methods in hybrid cognitive radio networks [6]. However, spectrum sensing is highly vulnerable to fading and hidden terminal problems between the PU and the SUs [7,8]. Thus, the decision of the spectrum sensing performed by a single SU is neither sufficient nor reliable for final decision about the presence of the PU in the network. To overcome the problem of the single SU sensing, researchers take advantages of cooperative spectrum sensing (CSS) for the enhancement of spectrum sensing. In CSS, each single SU gathers information about the PU channel, and shares its local sensing information with the FC, which accumulates the sensing information from the SUs and declares the global decision about the presence of the PU in the network [9,10]. In CSS, the SUs send information to the FC in two ways. In the first scenario, the SU sends a single bit of information to the FC, which is also known as the hard-decision rule. In the second scenario, which is called the soft combination rule, the SU sends a sampled energy value to the FC [11,12].

However, the existence of malicious users (MUs), or outliers, in the network highly degrades the performance of CSS. Various attacks, which highly degrade the performance of the networks, have been studied in the literature. Two common attacks are primary user emulation attacks (PUEAs) and spectrum sensing data falsification (SSDF) attacks [13,14]. In SSDF attacks, the MUs falsify the sensing results, which influences the sensing results in two ways. First, it decreases the probability of detection, which ultimately decreases the spectrum utilization. Second, it increases the probability of misdetection and the probability of false alarm, which increases interference in the network. Thus, overall performance of CSS is degraded by the SSDF attacks. To mitigate the effects of these attacks, several schemes have been proposed [15–17]. In Reference [18], the impact of incorrect information of the sensing system was formulated as detection performance and sensing efficiency; additionally, an authentication code length was proposed to reduce the system overhead. The authors of [19] proposed a MU suppression scheme, which consists of an improved energy detector followed by a statistical algorithm implemented at the FC. The authors of [20] proposed a neighbor detection-based spectrum sensing algorithm in distributed CRNs, which detects attackers with the help of neighbors during spectrum sensing to improve the decision-making accuracy. In this algorithm, the extreme outliers are isolated in the cognitive radio ad hoc network via the modified Z-test, and then the q-out-of-m rule is implemented to mitigate the SSDF attack [21]. Similarly, the authors integrated the reputation and q-out-of-m rule mechanism to mitigate the effect of the SSDF attack [22,23]. The authors of [24] utilized a k-medoids clustering algorithm to mine the collection of sensing reports at the FC to determine the attacker's presence; additionally, the proposed scheme can be utilized on streaming data (sensing reports), and thereby detects and isolates the attackers existing in the networks. The intelligent MUs were accurately detected by the authors of [25], who used a physical-layer network coding scheme based on a novel scheme friend or foe (FOF) detection. A cross-layered approach was presented to make the SU able to differentiate between the PU and MU through the hidden Markov model at the medium access control (MAC) layer [26]. The authors of [27] took advantage of the compressive sensing to detect the attack and defensive behavior and proposed a density-based MU detection with the trusted user to distinguish the MU precisely. A robust defense strategy against the MUs via double-sided neighbor distance-based genetic algorithm was presented in order to filter out the MU sensing reports in CSS [28]. The authors of [29] proposed a novel attacker identification algorithm that is able to skillfully detect attackers and reject their reported results. Moreover, a novel attacker punishment algorithm was provided with the aim of punishing attackers by lowering their individual energy efficiency, motivating them to quit sending false results. A comparative analysis of different outlier techniques was proposed by the authors of [30]. Similarly, a comparative analysis of the various outlier method for the MUs was discussed by the authors of [31]. The authors of [32] assigned a reputation value to the SU, while ignoring the SUs having a reputation below a threshold value. An extended sequential CSS scheme was proposed based on the

value of each SU [33]. A critical analysis of the MU attack, i.e., always yes, always no, and random attack, was studied by the authors of [34]. An onion peeling approach based on the calculation of suspicious levels was proposed by the authors of [35], which used belief propagation as the detection method. Protection of the CSS method, mentioned by the authors of [32], was reduced in the case of a large number of MUs.

In this paper, we propose a correlation-based scheme at the FC to detect the outlier and its behavior. In the proposed scheme, the FC first collects sensing information from all individual SUs, and then applies the correlation tool in the difference of the result of each SU, with the collective sensing results of all the SUs. The proposed scheme at the FC detects the results of the normal SUs, which are dissimilar from those of the MUs. In the proposed scheme, the box-whisker plot is utilized to classify the outlier and normal SUs. The box-whisker plot defines the upper and lower quartile limits of the normal SUs. Through the proposed scheme, the outlier and normal SUs are classified. The proposed scheme is tested for the existence of opposite malicious users (OMUs) and random opposite malicious users (ROMUs). The OMUs always send a high-energy signal when the PU is absent and a low-energy signal when PU is present. The ROMU is more dangerous and difficult to cope with. The ROMU's behavior is unpredictable, it behaves as a normal SU, while appearing as MU with opposite behavior at random intervals of time. Unlike always yes and always no, both the OMU and ROMU increase the probability of a false alarm and misdetection, which both degrades the bandwidth utilization and increases the interference to the PU network. Through the simulation study, we demonstrated that the proposed scheme can successfully classify the response of both OMUs and ROMUs from the normal SUs in a delicate manner.

The remaining sections of this paper are organized as follows. In Section 2, a detailed description of the system model is presented. In Section 3, we discuss the proposed scheme and describe the steps required for the classification of the outlier from the normal users. The performance evaluation and discussion are presented in Section 4. Finally, the paper is concluded in Section 5.

## 2. System Model

We considered a cooperative spectrum sensing scenario in a cognitive radio network. We assumed that the total number of outliers/MUs in the network was less than the total number of normal SUs. The system model for the proposed scheme is shown in Figure 1. The SUs performed spectrum sensing and sent the report to the FC, for the presence of the PU in the network. The SUs forwarded a hard-binary decision 1 if the spectrum was occupied by the PU, and -1 if the spectrum was not occupied by the PU. The FC received the local sensing reports from all SUs. The FC then employed the proposed scheme on these reports to identify the SU as an outlier on the basis of the history of each SU energy report. Once the outliers were identified and removed, the FC then employed a simple rule to declare a global decision about the presence of the PU in the network.

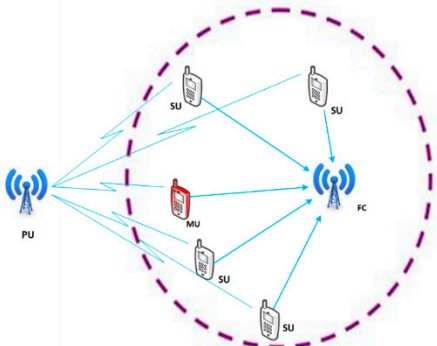

**Figure 1.** System model.

The information received at the receiver SU in a particular band for the presence or absence of the PU was represented as a binary hypothesis given as

$$
y_j(l) = \begin{cases} n_j(l) & ; \ H_0 \\ \\ h_j s(l) + n_j(l) & ; \ H_1 \end{cases}, \tag{1}
$$

where $H_0$ is the absence hypothesis, $H_1$ represents the presence hypothesis of the PU in the network, $y_j(l)$ shows the received signal from the $j$th SU, $n_j(l)$ is the additive white Gaussian noise (AWGN) in the $l$th time slot for the $j$th SU, $s(l)$ is the signal transmitted by the PU, and $h_j$ is the channel gain value between the PU and the SU in the $l$th time slot.

According to the hypotheses $H_0$ and $H_1$, the received signal energy of the channel by the $j$th SU at the $i$th sensing interval is

$$
E_j(i) = \begin{cases} \sum\limits_{l=l_i}^{l_i+K-1} |n_j(l)|^2 & ; \ H_0 \\ \\ \sum\limits_{l=l_i}^{l_i+K-1} |h_j s(l) + n_j(l)|^2 & ; \ H_1 \end{cases}, \tag{2}
$$

where $K$ denotes the number of samples in the $i$th sensing interval. According to central limit theorem (CLT), when the value of $K$ is large enough, then the energy reported by each SU converges to a Gaussian random variable under $H_0$ and $H_1$, which can be formulated as [28]:

$$
E_j \sim \begin{cases} N(\mu_0 = K, \sigma^2{}_0 = 2K) & ; \ H_0 \\ \\ N(\mu_1 = K(\eta_j + 1), \sigma^2{}_1 = 2K(\eta_j + 1)) & ; \ H_1 \end{cases}, \tag{3}
$$

where $\eta_j$ is the signal-to-noise ratio (SNR) between the $j$th SU and the PU, $(\mu_0, \sigma^2{}_0)$ is the mean and variance under $H_0$, and $(\mu_1, \sigma^2{}_1)$ are the mean and variance under $H_1$.

## 3. Proposed Scheme

In this paper, we proposed a correlation-based approach to identify the legitimate SUs and outliers. The box-whisker plot was introduced to classify the legitimate SUs from the outliers. In the proposed scheme, each SU senses the spectrum by utilizing the energy-detection technique and compares the received signal strength with a threshold value. On the basis of the sensing results, the SUs send a hard decision of 1 or $-1$ to the FC, which can be given as

$$
Z_j(i) = \begin{cases} 1 & ; \ if \ E_j(i) \geq \gamma_j \\ \\ -1 & ; \ Otherwise \end{cases}, \tag{4}
$$

where $E_j(i)$ is the energy received by the $j$th SU in the $i$th sensing interval, $\gamma_j$ is the value of the threshold set for the $j$th SU, and $Z_j(i)$ is the $j$th SU decision of the PU signal in the $i$th sensing interval, representing 1 if the $E_j(i)$ is greater than the threshold value and $-1$ if energy of the received signal $E_j(i)$ is smaller than the threshold value. The FC collects spectrum sensing information of the individual SU results with its own local decision as

$$
Z = \begin{bmatrix} z_{11} & z_{12} & z_{13} & z_{14} & \cdots & z_{1M} \\ z_{21} & z_{22} & z_{23} & z_{24} & \cdots & z_{2M} \\ z_{31} & z_{32} & z_{33} & z_{34} & \cdots & z_{3M} \\ \vdots & \vdots & \vdots & \vdots & \vdots & \vdots \\ z_{N1} & z_{N2} & z_{N3} & z_{N4} & \cdots & z_{NM} \end{bmatrix}, \tag{5}
$$

where $Z$ represents the sensing energy accumulated in the database of the FC by all the SUs' hard-decision values. In Equation (5), the rows represent the sensing intervals, and the columns represent the SUs' energy responses under each sensing interval. $M$ denotes the total number of SUs including the normal SUs, the outlier/MU, and the FC information, and $N$ is the number of sensing intervals. Furthermore, correlation was used as a tool for the detection of the most harmful and difficult detect OMU and ROMU users.

The correlation coefficients for the two samples $X$ and $Y$ can be determined as

$$r_{XY} = r_{XY=} \frac{\sum_p (X_p - \overline{X})(Y_p - \overline{Y})}{\sqrt{\sum_p (X_p - \overline{X})^2 \sum (Y_p - \overline{Y})^2}},$$
$$-1 \le r_{XY} \le +1 \tag{6}$$

where $\overline{X} = \frac{\sum_{p=1}^{N} X_p}{N}$ and $\overline{Y} = \frac{\sum_{p=1}^{N} Y_p}{N}$ are the mean values of the samples $X$ and $Y$, respectively, and $X_p$ and $Y_p$ are the $p$th elements of samples $X$ and $Y$, respectively. Equation (6) shows that the correlation of variable $X$ taken with $Y$ is the same as the correlation of $Y$ taken with $X$.

Correlation is a statistical exercise that shows how intensely the pair of testers are related to each other. Equation (6) shows the value of $r$ from $-1$, when both variables are in the opposite direction with a perfect negative correlation, to $+1$, when both variables are in the same direction with a perfect positive correlation. Effective use of this correlation process is a good measure of the relationship between the two variables when there is a chance of outliers, no normality, no steady variance, and nonlinearity existing between the two variables that are being examined.

### 3.1. Outlier Detection

All SUs send their sensing reports to the FC as shown in Equation (5). At the FC, a relationship is verified by comparing each SU's sensing decision with the other SUs, to determine any abnormal SU, which sends spectrum sensing falsification data to the FC. The FC is able to easily identify the both the OMU and ROMU category of outliers/MUs by the following three steps.

#### 3.1.1. Step One: Averaging Differences of the SUs

In this step, the FC determines the difference in the sensing results of the $j$th SU with the rest of the SUs. First, the average of all the SUs' sensing decisions is calculated by neglecting the $j$th SU's sensing result in the $i$th sensing interval to find the impact of excluding this particular SU in the overall sensing result. The same process is performed for all the $M$ SUs during each $N$th sensing interval to find the average of each SU in the FC, determined as

$$M = \begin{bmatrix} m_{11} & m_{12} & m_{13} & m_{14} & \cdots & m_{1M} \\ m_{21} & m_{2,2} & m_{23} & m_{24} & \cdots & m_{2M} \\ m_{31} & m_{32} & m_{33} & m_{34} & \cdots & m_{3M} \\ \vdots & \vdots & \vdots & \vdots & \vdots & \vdots \\ m_{N1} & m_{N2} & m_{N3} & m_{N4} & \cdots & m_{NM} \end{bmatrix},$$
$$M = [m_{ij}]$$
$$where \quad m_{ij} = \left\{ \frac{\left(\sum_{j=1}^{M} z_{i,j}\right) - z_{ij}}{M-1} \right\} \tag{7}$$

where $M$ is the total number of SUs, $N$ is the number of sensing intervals, $m_{ij}$ is the average value of the energy reports from all the other SUs in the $i$th sensing interval while ignoring the $j$th SU result. As the energy responses of both the OMU and ROMU are different from the rest, taking such outliers/MUs out of the average value calculation during each sensing interval by the FC generates a dissimilar result for the OMU and ROMU users compared with the normal SUs.

In order to estimate how much the individual sensing results of each SU, $z_{ij}$, are behaving differently from the average value of other users' results, $m_{ij}$, we considered the following

$$\Delta d_{ij} = z_{ij} - m_{ij}$$

$$\Delta d_{ij} = \begin{bmatrix} \Delta d_{11} & \Delta d_{12} & \ldots & \Delta d_{1M} \\ \Delta d_{21} & \Delta d_{22} & \ldots & \Delta d_{2M} \\ \ldots & \ldots & \ldots & \ldots \\ \Delta d_{N1} & \Delta d_{N2} & \ldots & \Delta d_{NM} \end{bmatrix}, \tag{8}$$

where $\Delta d_{ij}$ is the difference in the sensing results of the $j$th SU in the $i$th sensing interval, $z_{ij}$ is the individual sensing result of the $j$th SU in the $i$th sensing interval, and $m_{ij}$ is the average sensing result of the SUs other than the $j$th SU.

### 3.1.2. Step Two: SUs' Correlation

The FC measures the difference value of each SU with the rest of the SUs as in Equation (8). Once the difference between the SUs was determined, we utilized the correlation tool defined in Equation (6) for all SUs, which is measured as

$$rr_{\Delta d_j \Delta d_k} = r_{\Delta d_k \Delta d_j} = \frac{\sum_{i=1}^{N} \left( \Delta d_{ij} - \overline{\Delta d_j} \right) \left( \Delta d_{ik} - \overline{\Delta d_k} \right)}{\sqrt{\sum_{i=1}^{N} \left( \Delta d_{ij} - \overline{\Delta d_j} \right)^2 \sum_{i=1}^{N} \left( \Delta d_{ik} - \overline{\Delta d_k} \right)^2}}, \tag{9}$$

where $\Delta d_{ij}$ and $\Delta d_{ik}$ are elements of the $j$th and $k$th user sample in the $i$th sensing interval, and $\overline{\Delta d_j}$ and $\overline{\Delta d_k}$ are the mean values calculated for the $j$th and $k$th SU samples.

$$C = \begin{bmatrix} & SU_1 & \ldots & SU_M \\ SU_1 & r_{\Delta d_1 \Delta d_1} & \cdots & r_{\Delta d_1 \Delta d_M} \\ \ldots & \ldots & \ldots & \ldots \\ SU_M & r_{\Delta d_M \Delta d_1} & \cdots & r_{\Delta d_M \Delta d_2} \end{bmatrix}. \tag{10}$$

The data of the normal SUs are separated from those of the outliers or MUs by adding all the correlation differences in (10) for each SU as follows.

$$C_{ij} = \begin{bmatrix} \sum_{i=1}^{M} r_{\Delta d_i \Delta d_1} & \sum_{i=1}^{M} r_{\Delta d_i \Delta d_2} & \sum_{i=1}^{M} r_{\Delta d_i \Delta d_3} & \sum_{i=1}^{M} r_{\Delta d_i \Delta d_4} \cdots \sum_{i=1}^{M} r_{\Delta d_i \Delta d_M} \end{bmatrix}. \tag{11}$$

Based on the results of Equation (10), the OMU had more negative values, followed by the ROMU, when compared with normal SUs. From Equation (10), the behaviors of all three categories of SUs, i.e., normal SUs, OMUs, and ROMUs, were identified. These behaviors identify the outliers in the network. These outlier values were further identified and classified in step three of the proposed scheme.

### 3.1.3. Step Three: Outlier Classification Using the Box-Whisker Plot

In the proposed scheme, we utilized the box-whisker plot to find both the OMU and the ROMU as outliers in the result of Equation (11). A box-whisker plot divides Equation (11) into four parts. First, the results are made in order form and they are divided into an upper and lower half by the median. The median of the lower half is named as the lower quartile, while the median of the upper half is stated as the upper quartile. The lower and upper extremes are marked as the least and greatest values of the results. All the SUs' results of step two were arranged in ascending order from the lowest to the highest. The median value of the results can be calculated as $Med = median(C)$. The first quartile is denoted as $Q1_{Lower}$, which implies the value at the 25th percentile of $C$. The third quartile is also calculated as $Q3_{Lower}$, which implies the value at the 75th percentile of $C$, and thus, the inter-quartile value is determined by $IQR = Q3_{Lower} - Q1_{Lower}$. The upper and the

lower limits of the box-whisker plot were measured and marked for detection of the outlier values as follows:

$$Lower\ Limit = Q1_{Lower} - 1.5 * IQR. \tag{12}$$

$$Upper\ Limit = Q3_{Lower} + 1.5 * IQR. \tag{13}$$

Once the lower and the upper quartile limits were defined by the box-whisker plot by Equation (12) and Equation (13), an SU was declared as one of the outliers, i.e., OMU or ROMU, on the basis of the following criteria:

$$SU_j = \begin{cases} MU, & ;if\ Upper\ limit\ \leq Data \leq Lower\ Limit \\ Normal\ SU & ;Otherwise \end{cases}, \tag{14}$$

Since the outliers have different responses in Equation (11) than the normal SUs, they are classified as outliers with Equation (14). The overall flow chart of the proposed scheme is shown in Figure 2.

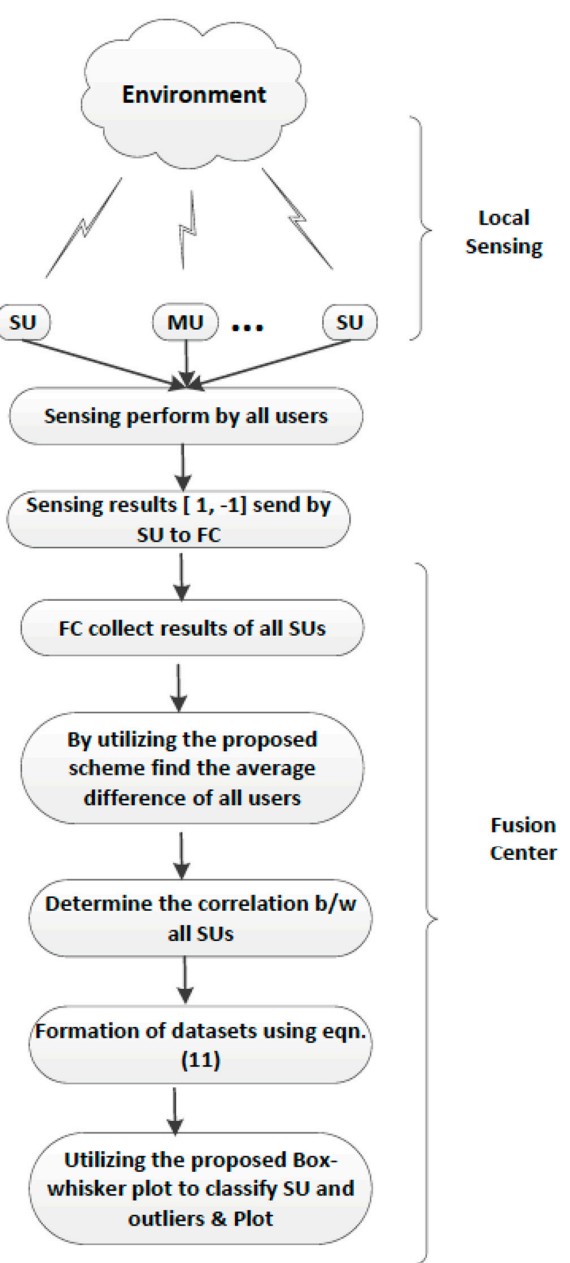

**Figure 2.** Flow chart of the proposed scheme.

## 4. Numerical Evaluation

In this section, we numerically evaluate the performance of the proposed scheme considering the parameters given in Table 1.

**Table 1.** Simulation parameters.

| Parameters | Values |
|---|---|
| Number of SUs | 10 |
| Number of OMUs | 1 |
| Number of ROMUs | 1 |
| Signal-to-noise ratio (SNR) [dB] | $-30$ to $-20$ |
| Number of iteration | 10,000 |
| Samples in each sensing interval | 270 |
| Sensing time | 1 ms |

To verify the effectiveness of the proposed scheme, we considered three scenarios. In the first scenario, we considered the only existence of the OMU with the normal SUs in the network. The OMUs are very sensitive to the performance of the network. Figure 3 shows the simulation results when the outlier behaves as an OMU. It can be observed that the normal SUs lie in the range of the lower and the upper quartile limits defined by the box-whisker plot, whereas the OMU lies outside the limits. These lower and upper quartiles define the boundary of the normal SUs in the network. Table 2 presents the values of the SNRs, the quartile, IQR and the lower and the upper quartile limits of the SUs in the network.

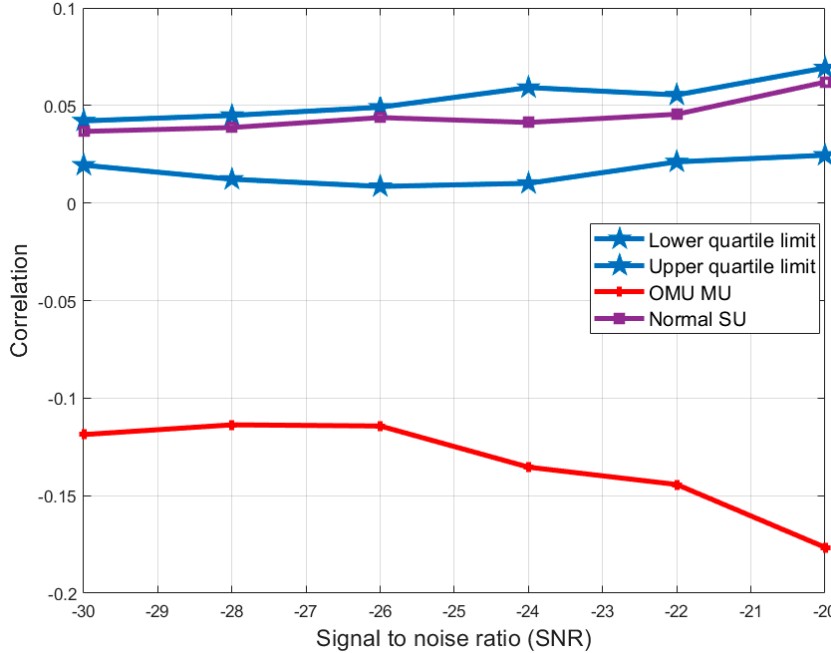

**Figure 3.** Correlation vs. signal-to-noise ratio (SNR), when an opposite malicious user (OMU) exists in the network.

In the second scenario, we considered only the existence of the ROMU and the normal SUs in the network. The ROMU behaves randomly with the probability *p* and appears as a normal SU with the probability 1-*p*. Figure 4 shows the simulation results in this scenario. The upper and lower quartile limits are defined in Table 3. It can be shown from Figure 4 that the box-whisker plot defines the limits of the lower and upper quartiles for the normal SUs and the proposed scheme can easily classify the normal SUs in this limit, whereas the ROMU response is different from the normal SUs, and easily identified in the system, which shows the effectiveness of the proposed scheme.

**Table 2.** Box-whisker plot data of the correlation results under OMU users only.

| SNR (dB) | Min | Q1 | Median | Q3 | Max | IQR | Lower Limit | Upper Limit |
|---|---|---|---|---|---|---|---|---|
| −30 | −0.11872 | 0.027923 | 0.029733 | 0.033575 | 0.036673 | 0.005652 | 0.019445 | 0.042053 |
| −28 | −0.11368 | 0.024431 | 0.029003 | 0.032603 | 0.038683 | 0.008172 | 0.012172 | 0.044861 |
| −26 | −0.1143 | 0.023746 | 0.026274 | 0.033884 | 0.043767 | 0.010139 | 0.008538 | 0.049092 |
| −24 | −0.13541 | 0.028492 | 0.035867 | 0.040777 | 0.041255 | 0.012285 | 0.010063 | 0.059205 |
| −22 | −0.14429 | 0.03397 | 0.040021 | 0.042547 | 0.045457 | 0.008577 | 0.021104 | 0.055413 |
| −20 | −0.17651 | 0.04133 | 0.048643 | 0.052537 | 0.061966 | 0.011206 | 0.024521 | 0.069346 |

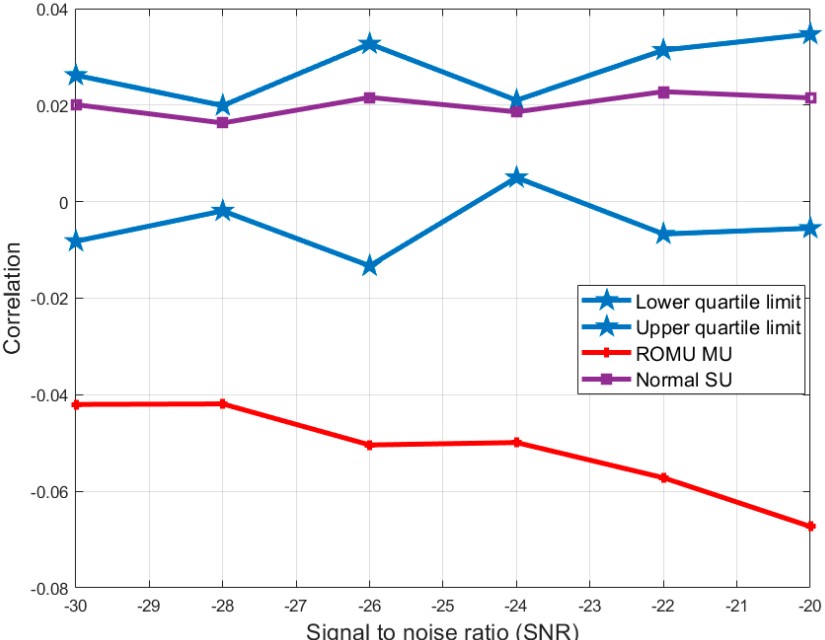

**Figure 4.** Correlation vs. SNR, when a random opposite malicious user (ROMU) exists in the network.

**Table 3.** Box-whisker plot data of the correlation results under ROMU users only.

| SNR (dB) | Min | Q1 | Median | Q3 | Max | IQR | Lower Limit | Upper Limit |
|---|---|---|---|---|---|---|---|---|
| −30 | −0.04204 | 0.004687 | 0.00771 | 0.013285 | 0.020143 | 0.008598 | −0.00821 | 0.026181 |
| −28 | −0.04186 | 0.006307 | 0.009428 | 0.011756 | 0.016341 | 0.005449 | −0.00187 | 0.019929 |
| −26 | −0.05041 | 0.003967 | 0.010962 | 0.01545 | 0.021564 | 0.011483 | −0.01326 | 0.032674 |
| −24 | −0.0499 | 0.01026 | 0.01173 | 0.013742 | 0.018565 | 0.003482 | 0.005036 | 0.018966 |
| −22 | −0.05722 | 0.007603 | 0.013448 | 0.017124 | 0.022818 | 0.009521 | −0.00668 | 0.031406 |
| −20 | −0.06726 | 0.009589 | 0.017555 | 0.019618 | 0.021469 | 0.010028 | −0.00545 | 0.03466 |

In the third scenario, we considered the existence of both the OMU and ROMU in the network. Table 4 defines the upper and the lower quartile limits for the normal SUs. Figure 5 shows the simulation results, when both the OMU and the ROMU are equally distributed. From Figure 5, we can observe that the normal SUs lie within the range of the upper and lower limits, whereas the OMU and ROMU are not in the range of the limits set by the box-whisker plot. Figure 5 shows that the detection results of the OMU were more negative compared to the ROMU. The ROMU behavior was closer to that of the normal SUs and more sensitive care was required for the detection of such outliers or of the MUs.

In Figure 6, we show the receiver operator characteristics (ROC) comparison of the proposed scheme with other existing schemes when the MU was present in the network, and when the MU did not exist in the network. Figure 6 demonstrates that when no scheme was applied, the probability of detection decreased and the probability of false alarm

and probability of misdetection increased. Furthermore, when the proposed scheme was applied in the presence of MU, the performance was better than other existing schemes.

**Table 4.** Box-whisker plot data of the correlation results under both OMU and ROMU users.

| SNR (dB) | Min | Q1 | Median | Q3 | Max | IQR | Lower Limit | Upper Limit |
|---|---|---|---|---|---|---|---|---|
| −30 | −0.1233 | 0.01559183 | 0.02097 | 0.023847249 | 0.025981888 | 0.008255 | 0.003209 | 0.03623 |
| −28 | −0.12418 | 0.01502566 | 0.02049 | 0.022928377 | 0.03319715 | 0.007903 | 0.003172 | 0.034782 |
| −26 | −0.12967 | 0.01767262 | 0.02225 | 0.0264105 | 0.029608583 | 0.008738 | 0.004566 | 0.039517 |
| −24 | −0.14605 | 0.02419445 | 0.02566 | 0.030521757 | 0.031995221 | 0.006327 | 0.014703 | 0.040013 |
| −22 | −0.15155 | 0.01600466 | 0.02709 | 0.031661185 | 0.035777566 | 0.015657 | −0.00748 | 0.055146 |
| −20 | −0.1768 | 0.0300924 | 0.03288 | 0.033382178 | 0.038434105 | 0.00329 | 0.025158 | 0.038317 |

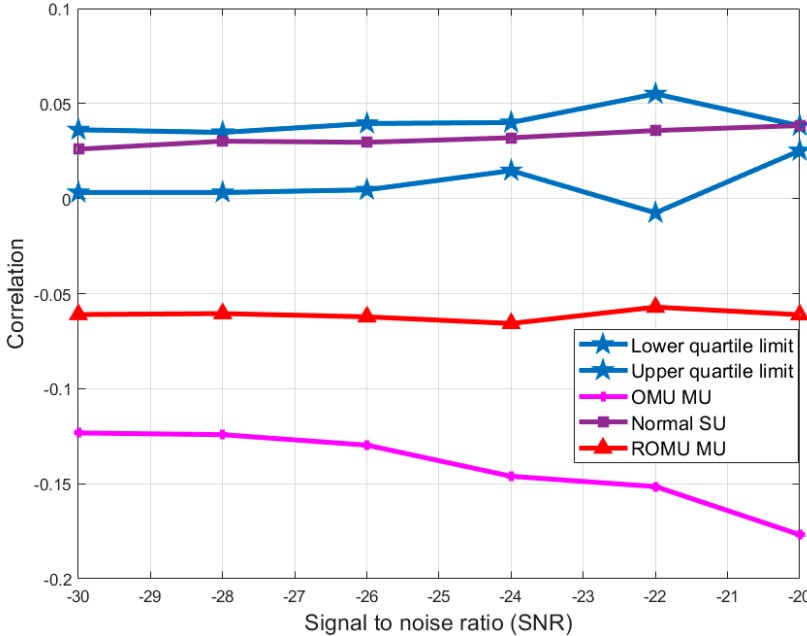

**Figure 5.** Correlation vs. SNR, when both the OMUs and ROMUs exist in the network.

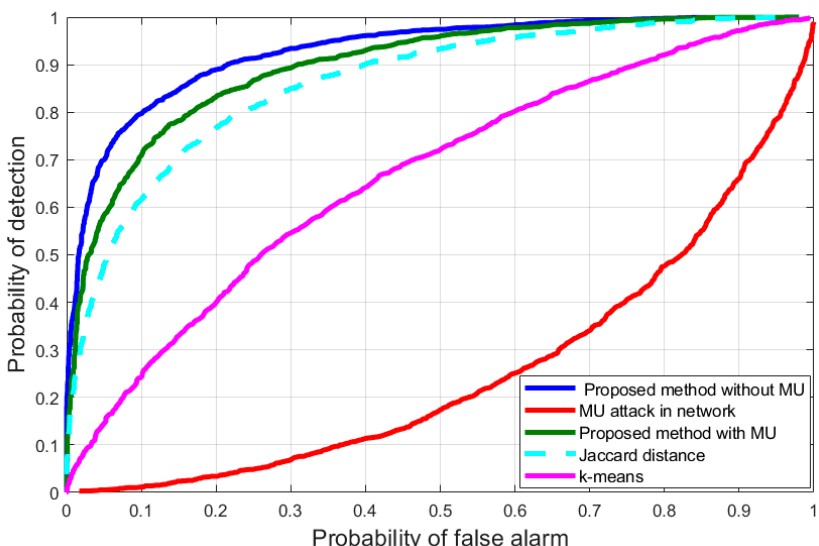

**Figure 6.** ROC curve of the proposed in comparison with other schemes.

Tabular and graphical results show that the proposed scheme was effective in detecting the outliers or MUs in CSS environments. The proposed scheme had the ability to identify

and classify both types of outliers, i.e., the OMU and the ROMU. By utilizing the box-whisker plot, an SU was classified as an outlier by the FC if its result lay above the upper quartile limit or below the lower quartile limit. Through the simulation results, we have shown that the proposed scheme can easily detect the outlier of OMU and ROMU in nature.

## 5. Conclusions

SSDF attacks severely degrade the performance of CRNs. In this paper, we proposed a correlation-based approach using the box-whisker plot for the detection of outliers in the networks. In the proposed scheme, we considered the hard decision of each SU, and the FC utilized correlation tools and calculated the correlation for finding the similarity of the sensing results of the SUs and outliers. By utilizing the correlation-based approach, we easily classified the outlier among the normal SUs. The outliers were further classified by using the box-whisker plot. The box-whisker plot defined the lower and upper quartile limits for the SUs. Finally, the normal user lay within the range, so the outliers were easily classified from the normal SUs. Through intensive simulation studies, we verified that the proposed scheme has the ability to classify the OMU and ROMU outliers in CRNs.

**Author Contributions:** All authors conceived and proposed the research idea. M.S.K. and M.F. designed the scenario; M.S.K. and S.A. performed the simulation results; M.S.-H. and S.M.K. analyzed the simulation results; M.S.K. and S.A. wrote the paper under the supervision of M.S.-H., S.M.K., and J.K. All authors have read and agreed to the published version of the manuscript.

**Funding:** This work was supported in part by the Ministry of Science and ICT (MIST), Korea, under the Information Technology Research Center (ITRC) support Program (IITP-2020-2018-0-01426) supervised by the IITP Institute for Information and Communication Technology Planning and Evaluation (IITP), and in part by the National Research Foundation (NRF), and funded by the Korea Government (MIT) (no.2019R1F1A1059125).

**Data Availability Statement:** The data used to support the finding of this study are included within article.

**Acknowledgments:** This work was supported in part by the Ministry of Science and ICT (MIST), Korea, under the Information Technology Research Center (ITRC) support Program (IITP-2020-2018-0-01426) supervised by the IITP Institute for Information and Communication Technology Planning and Evaluation (IITP), and in part by the National Research Foundation (NRF), and funded by the Korea Government (MIT) (no.2019R1F1A1059125).

**Conflicts of Interest:** The authors declare no conflict of interest.

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
