# Peer review of "A Correlation-Based Sensing Scheme for Outlier Detection in Cognitive Radio Networks"

_applsci, doi:10.3390/app11052362_

Round 1

Reviewer 1 Report

  1. The authors stated that "OMU and ROMU increases the probability of false alarm and mis-detection, while reducing bandwidth and increasing interference to the PU network. The reviewer suggest that simulation of probability of false alarm and mis-detection when OMU and ROMU exists in the network should be analyzed and discussed. 
  2. The references must be revised. There is lack of consistency. 

Reviewer 2 Report

My remarks are included in a separated file.

Reviewer 3 Report

The paper appears globally good. The introduction provides enough details about the state-of-art of malicious user detection in cognitive radio network. The introduced methodology is simple and quite well presented. You must modify Equation 14 where I think there is an error in MU interval definition. Even though you presented your results in a clean and easy to understand way, I believe that if you employ only a simulation methodology you should perform many more experiments. For example, you have not considered the case where more than 1 OMU/ROMU are involved. At the same time you should analyze the case where the number of SUs is much bigger.

Reviewer 4 Report

The manuscript attempts to address the malicious user's issue (outlier) behaving as a secondary user, thus degrading cognitive radio technology quality. The research topic discussed in the paper is a hot topic under wireless communication systems. The problem statement and proposed method are presented clearly in the manuscript. However, the manuscript requires a massive revision in its current form. Following are my comments:

  1. The manuscript contains numerous grammatical errors. I suggest the authors edit the paper with professional English editing institutions.
  2. In the Abstract, an abbreviation of ‘secondary user’ MUST be defined in its first appearance and use only the abbreviation term after that. Besides, any abbreviation defined in Abstract MUST also be defined again at other parts of the paper at their first use.
  3. From line 76 to 99, the term ‘In [ ]’ is used numerous times. It makes the paragraph monotonous. The authors should rewrite it.
  4. Lines 136, 137, and 138: Rewrite these sentences to make them meaningful.
  5. The symbols used in equations and their citation in the text should match their type (e.g., italics).
  6. How did the authors determine the ‘threshold value’ as mentioned in the proposed method?
  7. It is mentioned that SU reports the FC regarding the spectrum sensing. How does this happen? Which channel will be used for this communication? A full description of FC is required.
  8. Is the equation (10) correct?
  9. All the figures can be drawn in a more presentable form, especially the removal of blur is required.
  10. How did the authors determine the values of the parameters in Table 1? What happens when the number of SU is increased or decreased?
  11. The paper lacks a fair comparison of their proposed method with the existing methods. Without a fair comparison, how can authors say that their method is effective?
  12. The references MUST be updated with the latest literature.

Round 2

Reviewer 1 Report

The reviewer is satisfied with the revised version of the manuscript.

Reviewer 4 Report

The authors have addressed my comments. I recommend accepting the paper. 
